# FedEve: On Bridging the Client Drift and Period Drift for Cross-device Federated Learning

## Abstract

Federated learning (FL) is a machine learning paradigm that allows multiple clients to collaboratively train a shared model without exposing their private data. Data heterogeneity is a fundamental challenge in FL, which can result in poor convergence and performance degradation. *Client drift* has been recognized as one of the factors contributing to this issue resulting from the multiple local updates in FedAvg . However, in cross-device FL, a different form of drift arises due to the partial client participation, but it has not been studied well. This drift, we referred as *period drift*, occurs as participating clients at each communication round may exhibit distinct data distribution that deviates from that of all clients. It could be more harmful than client drift since the optimization objective shifts with every round. In this paper, we investigate the interaction between period drift and client drift, finding that period drift can have a particularly detrimental effect on cross-device FL as the degree of data heterogeneity increases. To tackle these issues, we propose a predict-observe framework and present an instantiated method, FedEve , where these two types of drift can compensate each other to mitigate their overall impact. We provide theoretical evidence that our approach can reduce the variance of model updates. Extensive experiments demonstrate that our method outperforms alternatives on non-iid data in cross-device settings.

## 1 Introduction

Federated learning is a decentralized machine learning approach that enables multiple clients to collaboratively train a shared model without exposing their private data (McMahan et al., 2017). In this paradigm, each client independently trains a local model using its own data and subsequently sends the model updates to a central server. The server then periodically aggregates these updates to improve the global model until it reaches convergence. There are two primary settings in FL: cross-silo and cross-device (Kairouz et al., 2021). Cross-silo FL typically involves large organizations (small number of clients), where most clients actively participate in every round of training (Chen and Chao, 2021; Lin et al., 2020). In contrast, cross-device FL focuses on scenarios like smartphones (huge number of clients, e.g., millions), where only a limited number of clients participate in each round (Li et al., 2020b; Reddi et al., 2020), due to communication bandwidth, client availability, and other issues. This paper primarily focuses on the cross-device setting with partial client participation since we discover and then solve its unique challenge —"***period drift***".

Distinguished from traditional distributed optimization, the statistical heterogeneity of data has been acknowledged as a fundamental challenge in FL (Li et al., 2020a; Chen and Chao, 2021; Lin et al., 2020). This data heterogeneity refers to the violation of the independent and identically distributed (non-iid) data assumption across clients, which can result in poor convergence and performance degradation when using FedAvg . *Client drift* is recognized as one of the factors contributing to this issue and attracts numerous efforts to address it (Karimireddy et al., 2021; Li et al., 2020b; Reddi et al., 2020). This phenomenon is characterized by clients who, after multiple local updates, progress too far towards minimizing their local objective, consequently diverging from the shared direction. However, in cross-device FL, a different form of drift exists and could be more detrimental to the training process than client drift, which has not been extensively studied. *This drift occurs periodically as different clients participate in each communication round, and these participating clients as a group may exhibit distinct data distribution that deviates from the overall distribution of all clients.*

This deviation could potentially lead to slow and unstable convergence, as the optimization objective shifts with every round. For simplicity, we refer to this phenomenon as ***period drift***. Despite both period drift and client drift being rooted in data heterogeneity, they stem from different causes (as illustrated in Figure 1). Client drift results from multiple local up-

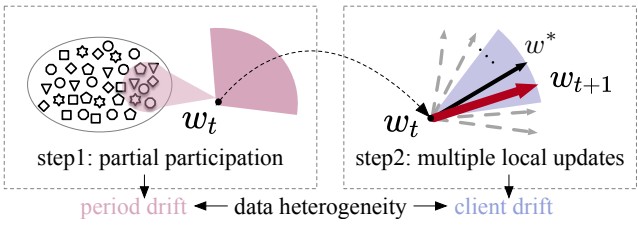

Figure 1: **The generation of period drift and client drift.**

dates and the non-iid, while period drift arises due to partial client participation and the non-iid. The combined effect of period drift and client drift further complicates the process of reaching stable and efficient convergence and makes it more challenging in cross-device FL.

In this paper, we first investigate the impact of period drift and client drift, finding that period drift can have a particularly detrimental effect on cross-device FL as the degree of data heterogeneity increases (as demonstrated in detail in Section 4.2). *While the impacts of period drift and client drift are additive, we fortunately uncover a cooperative mechanism therby these two types of drift can compensate each other to mitigate their overall impact.* To achieve this, we propose a predict-observe framework, where we consider at each round 1) the server optimization (e.g., momentum) as a prediction of a update step of FL; 2) the clients' optimization (e.g., local SGD) as an observation of this update step. Note that the vanilla FEDAVG is a special case in which the server does not make any predictions and solely relies on the observation provided by clients. In this framework, period drift and client drift are viewed as the noise respectively associated with prediction and observation. We thereby incorporate a Bayesian filter to integrate prediction (with period drift) and observation (with client drift) to achieve a better estimation of update step and reduce uncertainties. Based on the predict-observe framework, we present an instantiated method, referred as FEDEVE , which combines the prediction and observation through linear interpolation. The coefficient of this linear combination indicate the relative confidence between prediction and observation, which is determined by the variance of the period drift and client drift, thus produces a more precise estimation of updates. FEDEVE does not increase the client storage or extra communication costs, and does not introduce additional hyperparameter tunning, making it ideal for cross-device FL.

**Contributions**  We summarize the primary contributions of this paper as follows:

- We analyze the impact of period drift and client drift for cross-device FL, and observe that period drift has a particularly detrimental effect as the degree of data heterogeneity increases.
- We propose a predict-observe framework for cross-device FL that incorporates a Bayesian filter to integrate server optimization and clients' optimization so that period drift and client drift can compensate for each other.
- As an instantiation of the proposed framework, we present FEDEVE to combine prediction and observation through linear interpolation based on the variance of the period drift and client drift.
- We provide theoretical evidence within our framework that FEDEVE can reduce the variance of model updates. Extensive experiments demonstrate that our method outperforms alternatives on non-iid data in cross-device settings. [1]

## 2 RELATED WORKS

There are many works that have attempted to address the non-iid problem in federated learning. FEDAVG , first presented by McMahan et al. (2017), has been demonstrated to have issues with convergence when working with non-iid data. Zhao et al. (2018) depict the non-iid trap as weight divergence, and it can be reduced by sharing a small set of data. However, in traditional federal setting, data sharing violates the principle of data privacy. Karimireddy et al. (2021) highlight the phenomenon of "client drift" that occurs when data is heterogeneous (non-iid), and uses control variates to address this problem. However, using Scaffold in cross-device FL may not be effective, as it requires clients to maintain the control variates, which may become outdated and negatively impact performance. Li et al. (2020b) propose FedProx that utilizes a proximal term to deal with heterogeneity.

---

[1]For a smoother reading experience, please feel free to check out our reading guide in Appendix A.

In addition to these works, some research has noticed the presence of period drift, but have not specifically addressed it in their analysis. For example, Cho et al. (2022); Fraboni et al. (2023) investigate the problem of biased client sampling and proposes an sampling strategy that selects clients with large loss. However, active client sampling can potentially alter the overall data distribution by having unrandom clients participation, which can raise concerns about fairness. Similarly, Yao et al. (2019) propose a meta-learning based method for unbiased aggregation, but it requires training the global model on a proxy dataset, which may not be feasible in certain scenarios where such a dataset is not available. Zhu et al. (2022) observe that the data on clients have periodically shifting distributions that changed with the time of day, and model it using a mixture of distributions that gradually shifted between daytime and nighttime modes. Guo et al. (2021) study the impact of time-evolving heterogeneous data in real-world scenarios, and solve it in a framework of continual learning. Although these two papers define similar terms, they focus on the case of client data changing over time. However, in this paper, we find that even if the distribution of client data remains unchanged, period drift can seriously affect the convergence of FL.

## 3 METHODOLOY

In this section, we discuss the problem of some methods (e.g., FEDAVG ) in cross-device FL, and then propose our predict-observe framework and a method FEDEVE to deal with it.

### 3.1 TYPICAL FEDERATED LEARNING SETUP

Federated learning, as described by McMahan et al. (2017), involves utilizing multiple clients and a central server to optimize the overall learning objective. The goal is to minimize the following objective function:

$$\min_{w} f(w) = \sum_{k=1}^{N} p_k F_k(w) = \mathbb{E}_k[F_k(w)], \tag{1}$$

where $N$ is the number of clients, $p_k \geq 0$, and $\sum_k p_k = 1$. In general, the global objective is the expectation of the local objective over different data distributions $\mathcal{D}_k$, i.e., $F_k(w) = \mathbb{E}_{x_k \sim \mathcal{D}_k} f_k(w; x_k)$, with $n_k$ samples on each client $k$ and weighted by $p_k$. We set $p_k = \frac{n_k}{n}$, where $n = \sum_k n_k$ is the total number of data points. In deep learning setting, $F_k(w)$ is often non-convex. A common approach to solve the objective (1) in federated settings is FEDAVG (McMahan et al., 2017). For example, in cross-device FL, a small subset $\mathcal{S}_t$ ($|\mathcal{S}_t| \ll N$) of the total clients are selected at each round (ideally randomly, but possibly biased in practice), and then the server broadcasts its global model to the selected client. In parallel, each of the selected clients runs SGD on their own loss function $F_k(\cdot)$ for $E$ number of epochs, and sends the resulting model to the server. The server then updates its global model as the average of these local models and repeats this process until convergence.

One problem of FL is the non-iid data across clients, which can bring about "client drift" in the updates of each client, resulting in slow and unstable convergence (Karimireddy et al., 2021). Despite efforts to address the problem of client drift (Karimireddy et al., 2021; Li et al., 2020b; Reddi et al., 2020), there is a lack of research on the issue of period drift, i.e. the data distribution of selected clients at each round may differ from the overall data distribution of all clients. Period drift along with client drift can greatly impact the convergence of the learning process in FL, thus we propose a predict-observe framework to deal with them.

### 3.2 THE IMPACT OF DRIFT

In contrast to conventional distributed optimization, federated learning possesses distinct characteristics, such as client sampling, multiple local epochs, and non-iid data distribution. These attributes may lead to a drift in the updates of global model, resulting in suboptimal performance. This drift can be thought of as a noise term that is added to the true optimization states during the optimization process. Thus, we can make the assumption as:

**Assumption 3.1.** *The aggregated model parameters on the server $w_{server}$, can be represented as the sum of the optimal parameters $w^*$ and a drift (noise) that follows a normal distribution $w_{drift} \sim \mathcal{N}(0, \sigma_{drift}^2)$:*

$$w_{server} = w^* + w_{drift} \leftarrow noise, \tag{2}$$

where $w^*$ represents the optimal parameters obtained through the use of stochastic gradient descent (SGD), $w_{drift}$ represents the noise term caused by factors such as client sampling, multiple

local epochs, and non-iid data distribution that we assume a normal distribution, and $w_{server}$ represents the aggregated model parameters also follows a normal distribution $w_{server} \sim \mathcal{N}(w^*, \sigma^2_{drift})$, with the expectation of the aggregate model parameters being equal to the optimal parameters, i.e. $\mathbb{E}[w_{server}] = w^*$. Note that the assumption of Gaussian-like noise is natural, and its justification can be found in Appendix A.2.

In order to investigate the effect of the deviation on performance in FL, we utilize a regression optimization objective as in previous studies, such as (Zhang et al., 2019) and (Wu et al., 2018):

$$\hat{\mathcal{L}}(w) = \frac{1}{2}(w + w_{drift})^T A(w + w_{drift}),$$

where $w_{drift} \sim \mathcal{N}(0, \sigma^2)$ is the drift caused by the characteristics of FL. Therefore, the generalization error can be formulated as:

$$\mathcal{L}\left(w^t\right) = \mathbb{E}\left[\hat{\mathcal{L}}\left(w^t\right)\right] = \frac{1}{2}\mathbb{E}\left[\sum_i a_i \left(w_i^{t^2} + \sigma_i^2\right)\right]$$

$$= \frac{1}{2}\sum_i a_i \left(\mathbb{E}\left[w_i^t\right]^2 + \mathbb{V}\left[w_i^t\right] + \sigma_i^2\right).$$

In the context of FL, the generalization error can also be decomposed into three components: bias, variance, and noise. The noise component in this context is further influenced by factors such as client sampling, multiple local epochs, and non-iid data distribution, leading to a much larger overall generalization error compared to traditional stochastic gradient descent. Thus, our goal is to reduce the variance of drift $\sigma^2$ in order to improve both the convergence and performance of the model. By reducing the variance of drift, we can ensure that the updates made to the model are more consistent and accurate, leading to better overall performance. The subsequent section of this study aims to investigate the influence of drift on both the server and client side with respect to this noise component in federated learning.

### 3.3 THE PREDICT-OBSERVE FRAMEWORK

Initially, we establish the concept of period drift, represented by $Q_t$, and client drift, represented by $R_t$ at the $t$-th communication round. We first make an assumption of independence concerning the two types of drift, which states that the two drifts are independent of one another. This assumption allows us to more accurately analyze the impact of each drift on the model's performance and devise methods to mitigate their effects.

**Assumption 3.2.** *The initialization model parameters are independent of all period drifts $Q_t$ and client drifts $R_t$ at each communication round, that is $w_0 \perp Q_0, Q_1, \cdots, Q_t$ and $w_0 \perp R_0, R_1, \cdots, R_t$.*

The justification and limitation of this assumption can be found in Appendix A.3. Since the clients participating in each round in cross-device FL is only a small fraction of all clients, period drift can be attributed to the discrepancy that the objective of selected clients at each round does not align with the overall objective. Thus, an effective prediction of updates can potentially help reduce the period drift. As formulated in Equation (2), we express the prediction of updates on the server as:

$$\hat{w}_{t+1} = g(w_t) + Q_t, \quad Q_t \sim \mathcal{N}(0, \sigma^2_{Q_t}), \tag{3}$$

where $\hat{w}_{t+1}$ is the prediction model of $(t+1)$-th round as the output of predcit function $g(\cdot)$ with the current model $w_t$ as input. It is noteworthy that the period drift at the $t$-th round is represented by $Q_t$, and just like the drift in assumption 3.1, it is assumed to follow a normal distribution $\mathcal{N}(0, \sigma^2_{Q_t})$, characterized by a mean of zero and a variance of $\sigma^2_{Q_t}$. Client drift can be attributed to the phenomenon that the averaged optima of objectives does not align with the optima of averaged objectives. Thus, we consider the updates provided by these clients is a kind of observation of global updates. As formulated in Equation (2), we express it as:

$$\tilde{w}_{t+1} = h(\hat{w}_{t+1}) + R_t, \quad R_t \sim \mathcal{N}(0, \sigma^2_{R_t}), \tag{4}$$

where $\tilde{w}_{t+1}$ is the model of $(t+1)$-th round as the output of observe function $h(\cdot)$ with the predict model $w_t$ as input. Also, the client drift at the $t$-th round is represented by $R_t$, and just like the drift in assumption 3.1, it is assumed to follow a normal distribution $\mathcal{N}(0, \sigma^2_{R_t})$, characterized by a mean of zero and a variance of $\sigma^2_{R_t}$. It is clear that standard FEDAVG is a special case since there is no prediction for server optimization, and it solely relies on the observations provided by clients. Furthermore, the period drift, $Q_t$, and the client drift, $R_t$, are represented as noise terms that are incorporated into the prediction and observation functions. According to assumption (3.2), these drifts are independent of the current model states, and the lemma of independence noise is posited:

**Lemma 3.3.** *(Independence of Noise). the noise present in the prediction and observation at each communication round is independent of the current model state, specifically, $w_t \perp Q_t$ and $w_t \perp R_t$.*

The complete proof of the independence of noise can be found in appendix A.4. The equations presented in equations 3 and 4 depict the prediction and observation of updates, respectively, taking into account both period drift and client drift. In order to reconcile the discrepancy between the prediction (including period drift) and observation (including client drift), a Bayesian filter is introduced to allow for compensation between the two sources of drift. The prior probability of $w_{t+1}$ is represented by $P(\hat{w}_{t+1})$, and by combining the observation $P(\tilde{w}_{t+1})$ and the likelihood $P(\tilde{w}_{t+1} \mid \hat{w}_{t+1})$, the posterior probability $P(w_{t+1} \mid \tilde{w}_{t+1})$ of $w_{t+1}$ can be calculated as the new model at the $(t+1)$-th round, as shown in Equation (5).

$$P(\ w_{t+1}\ ) := P(\ \hat{w}_{t+1}\ \mid\ \tilde{w}_{t+1}\ ) = \frac{P(\ \tilde{w}_{t+1}\ \mid\ \hat{w}_{t+1}\ )P(\ \hat{w}_{t+1}\ )}{P(\ \tilde{w}_{t+1}\ )}. \tag{5}$$

By utilizing the Bayesian filter in our predict-observe framework, an update mechanism is implemented that first performs prediction and then observes the predicted model state, as described in the following procedure:

$$f^+_{w_t}(w) \overset{\text{predict}}{\Longrightarrow} f^-_{\hat{w}_{t+1}}(w) = \int_{-\infty}^{+\infty} f_{Q_t}[w - f(v)] f^+_{w_t}(v)\mathrm{d}v$$

$$\overset{\text{observe}}{\Longrightarrow} f^+_{w_{t+1}}(w) = \eta_t \cdot f_{R_t}[w_{t+1} - h(w)] \cdot f^-_{\hat{w}_{t+1}}(w), \tag{6}$$

where $f^+_{w_t}(w)$ is the posterior probability of $w_t$, $f^-_{\hat{w}_{t+1}}(w)$ is the prior probability of $w_{t+1}$, $f_{Q_t}$ is the PDF of period drift, $f^+_{w_{t+1}}(w)$ is the posterior probability of $w_{t+1}$, $f_{R_t}$ is the PDF of client drift, and $\eta_t = \left\{ \int_{-\infty}^{+\infty} f_{R_t}[\tilde{w}_{t+1} - h(\hat{w}_{t+1})] f^-_{\hat{w}_{t+1}}(w)\mathrm{d}w \right\}^{-1}$. By combining prediction and observation, the fused model can be estimated by taking the expectation of the posterior probability as follow:

$$\hat{w}_{t+1} = E\left[f^+_{w_{t+1}}(w)\right] = \int_{-\infty}^{+\infty} w f^+_{w_{t+1}}(w)\mathrm{d}w. \tag{7}$$

**Theorem 3.4.** *Given assumption 3.1 and lemma 3.3, the composite model will exhibit a diminished degree of variance in comparison to the individual variances of both period drift and client drift, and the mean will be a linear combination that is weighted by the variances:*

$$\mu_{\text{fused}} = \frac{\mu_1 \sigma^2_{R_t} + \mu_2 \hat{\sigma}^2_{t+1}}{\sigma^2_{R_t}},$$

$$\sigma^2_{\text{fused}} = \frac{\hat{\sigma}^2_{t+1} \sigma^2_{R_t}}{\hat{\sigma}^2_{t+1} + \sigma^2_{R_t}}, \tag{8}$$

*where $\mu_1, \mu_2, \mu_{fused}$ is the mean of prediction, observation and fused model, and $\hat{\sigma}^2_{t+1}, \sigma^2_{R_t}, \sigma^2_{fused}$ is the variance of prediction, client drift and fused model.*

The complete proof of the bayesian filter can be found in appendix A.5. The application of Bayesian filtering allows for the interaction of period drift and client drift to generate a new model, which is characterized by a reduced level of variance as compared to the individual variances of period drift and client drift, as depicted in Figure 2(a). However, the computation of the new model is challenging due to the presence of infinite integrals in Equation (7) and $\eta_t$, as it is a general framework for any prediction and observation function. In the following section, we will propose a specialized method to facilitate the convergence of FL.

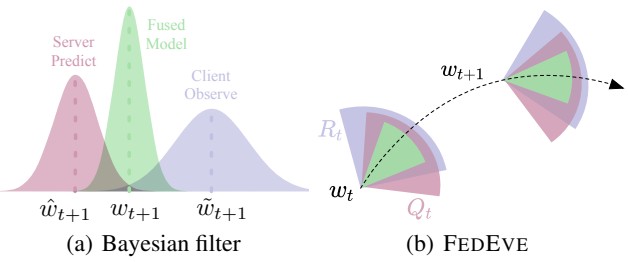

(a) Bayesian filter      (b) FEDEVE

Figure 2: **Illustrations of the framework and FEDEVE** .

### 3.4 THE FEDEVE METHOD

The predict-observe framework has been proposed as a strategy for mitigating the challenges of period drift and client drift. However, it also raises some questions regarding the effective method of prediction and the variance associated with both period and client drift. In this section, we demonstrate that the utilization of momentum as a server optimization (Hsu et al., 2019; Reddi et al., 2020), can serve as an effective prediction method. Furthermore, we present a method for estimating the variance of period drift and client drift. In the context of the predict-observe framework, we have adapted it to a specific setting where Nesterov momentum is employed as the prediction function $g(\cdot)$, and the observation function $h(\cdot)$ is the average of the models from the clients the same as FEDAVG . We have reformulated FEDAVG in an incremental form as the starting point of our approach.

$$\begin{aligned} w_{t+1} = \sum_{k \in \mathcal{S}_t} p_k w_t^k = w_t - \sum_{k \in \mathcal{S}_t} p_k \left( w_t - w_t^k \right) \\ = w_t - \sum_{k \in \mathcal{S}_t} p_k \Delta w_t^k = w_t - \Delta w_t. \end{aligned} \tag{9}$$

This formulation facilitates the accumulation of $\Delta w_t$ as the momentum on the server, which serves as a prediction of updates, as the empirical value of the hyperparameter $\beta = 0.9$ suggests that the direction of historical updates is likely to be maintained. By introducing the Nesterov momentum and specialize $g(w_t) = w_t - \eta_g M_t$ in Equation (3) as the prediction function. Additionally, we specialize $h(\hat{w}_{t+1}) = \hat{w}_{t+1} - \eta_g \Delta \tilde{w}_t$ in Equation (4) as the observation function. Thus, the predict-observe equation can be rewritten as follow:

$$\hat{w}_{t+1} = w_t - \eta_g M_t + Q_t, \tag{10}$$
$$\tilde{w}_{t+1} = \hat{w}_{t+1} - \eta_g \Delta \tilde{w}_t + R_t, \tag{11}$$

where $M_t$ is the momentum (the accumulation of $\Delta w_t$) at $t$-th round, $\Delta \tilde{w}_t$ is the average of model update in Equation (9) from clients at the states of $\hat{w}_{t+1}$, and $\eta_g$ is the global learning rate. By assuming a normal distribution for $Q_t$ and $R_t$ based on the equations (3), (4), and (5), the problem of infinite integral in Equation (7) and $\eta_t$ can be solved in a closed-form, as detailed in reference A.5.2. Additionally, due to the normal distribution, the form of distribution like equations 6, 7 is not necessary, and only the mean and variance are used to depict the model update process. Since these equations are linear in nature, the Bayesian filter can be specialized as the Kalman Filter (KF). The process of model update can thus be summarized as the use of KF, as represented by the following formulation:

$$\hat{w}_{t+1} = w_t - \eta_g M_t, \tag{12a}$$
$$\hat{\sigma}_{t+1}^2 = \sigma_t^2 + \sigma_{Q_t}^2, \tag{12b}$$
$$K = \frac{\hat{\sigma}_{t+1}^2}{\hat{\sigma}_{t+1}^2 + \sigma_{R_t}^2}, \tag{12c}$$
$$M_{t+1} = M_t + K(\Delta \tilde{w}_t - M_t), \tag{12d}$$
$$w_{t+1} = w_t - \eta_g M_{t+1}, \tag{12e}$$
$$\sigma_{t+1}^2 = (1 - K)\hat{\sigma}_{t+1}^2. \tag{12f}$$

The six steps of model update for each communication round in our method are outlined in Equations (12a)-(12f). Equation (12a) predicts the model states $w_t$ using the momentum $M_t$. Equation (12b) estimates the variance of the prediction model by summing the variance of $w_t$ and the period drift $Q_t$. To provide a clear representation, the variance of the prediction model is represented by $\sigma^-$ and the variance of the fused model is represented by $\sigma^+$. The core of our method is presented in Equation (12c), where the Kalman gain $K$ is calculated based on the ratio of the variance of the prediction $\hat{\sigma}_{t+1}^2$ and the observation (client drift) $R_t$.

The value of $K$ determines the relative weight of the prediction and observation when they are combined. Equation (12d) fuses the prediction and observation in a linear fashion, weighted by the Kalman gain $K$ calculated in (12c). The fourth line updates the global model with the fused $M_{t+1}$ calculated in (12d). Equation (12e) estimates the variance of the fused model $w_{t+1}$ using $K$ in (12d) and $\hat{\sigma}_{t+1}^2$ in (12b), which will be used in the next communication round. It is worth noting that all these calculations are performed on the server, thus our method retains the same level of communication cost as FEDAVG while also being compatible with cross-device FL settings.

While Equations (12a)-(12f) provide an efficient and accurate method for model updates, the variance of the period drift $\sigma_{Q_t}^2$ in Equation (12b) and the client drift $\sigma_{R_t}^2$ in Equation (12c) remains

unresolved. To address this issue, we propose an effective method for estimating the variance of the period drift and client drift.The period drift, which is a measure of the deviation from the consistency of the optimization objective at each communication round, can be quantified by analyzing the discrepancy between the prediction and the observation. Specifically, this can be done by computing the variance between the momentum $M_t$ and the average of the model updates $\Delta \tilde{w}_t$. Similarly, the client drift, which represents the inconsistency of the updates made by different clients, can be estimated by computing the variance between the average of the model updates $\Delta \tilde{w}_t$ and the updates made by each individual client $\Delta \tilde{w}_t^k$. We formulate the estimation of the variance of period drift and client drift as follows:

$$\sigma_{Q_t}^2 := \frac{\sum_{i=1}^d (M_t^i - \Delta \tilde{w}_t^i)^2}{|\mathcal{S}_t| d},$$

$$\sigma_{R_t}^2 := \frac{\sum_{k \in \mathcal{S}_t} \sum_{i=1}^d (\Delta \tilde{w}_t^{k,i} - \Delta \tilde{w}_t^i)^2}{|\mathcal{S}_t|^2 d},$$

(13)

where the index of model parameters is represented by the uppercase $i$ and the dimension of the model is represented by $d$. With the estimation of the variance of period drift and client drift, the overall process of model update can be described in Algorithm 1.

The fundamental principle of FEDEVE is to calculate the Kalman gain $K$ which is used to determine the relative weight of the prediction and observation when they are combined. The value of K is calculated based on the ratio of the variance of the prediction $\sigma_{Q_t}^2$ and the variance of the observation $\sigma_{R_t}^2$. This coefficient is used to adjust the update direction of the model. A small $K$ means that the observation is close to the prediction, hence the update direction will also be close to the prediction. A large $K$ means that the observation deviates significantly from the prediction, hence the update direction will deviate from the prediction and be closer to the observation. This allows the algorithm to adapt to different scenarios in which the observations may deviate more or less from the predictions.

## 4 EXPERIMENTS

### 4.1 SETUP

**Datasets and models.**  We evaluate FEDEVE on three computer vision (CV) and recommender system (RS) datasets under realistic cross-device FL settings. *For CV dataset*, we use FEMNIST[2] Caldas et al. (2018), consisting of 671,585 training examples and 77,483 test samples of 62 different classes including 10 digits, 26 lowercase and 26 uppercase images with 28x28 pixels, handwritten by 3400 users. We also use CIFAR-10/100 [3] Caldas et al. (2018), consisting of 50,000 training examples and 10,000 test samples of 10/100 different classes with 32x32 pixels. For FEMNIST dataset, we use the lightweight model LeNet5 LeCun et al. (1998) and for CIFAR-10/100 dataset, we use ResNet-18 (replacing batch norm with group norm (Hsieh et al., 2020; Reddi et al., 2020)). *For RS dataset*, we use MovieLens 1M [4]Harper and Konstan (2015), including 1,000,209 ratings by unidentifiable 6,040 users on 3,706 movies. It is a click-through rate (CTR) task, and we use the popular DIN Zhou et al. (2018) model. For performance evaluation, we follow a widely used leave-one-out protocol Muhammad et al. (2020). For each user, we hold out their latest interaction as testset and use the remaining data as trainset, and binarize the user feedback where all ratings are converted to 1, and negative instances are sampled 4:1 for training and 99:1 for test times the number of positive ones.

**Federated learning settings.**  It is important to note that the datasets FEMNIST and MovieLens 1M have a "natural" non-iid distribution, which means that the data is split by "user_id". For example, in FEMNIST, images are handwritten by different users, and in MovieLens 1M, movies are rated by different users. Furthermore, we use the Dirichlet distribution, to simulate the label distribution skew setting for FEMNIST, as described in Hsu et al. (2019). This distribution allows us to control the degree of heterogeneity by adjusting the hyperparameter $\alpha$ (the smaller, the more non-iid). This allows us to test the robustness of the algorithm under different levels of heterogeneity, which is a common scenario in real-world FL settings. For the FL training, we set a total of $T = 1500$ communication rounds for the CV task and sample 10 clients per round with SGD optimizer. For the

---

[2]https://github.com/TalwalkarLab/leaf/tree/master/data/femnist

[3]https://www.cs.toronto.edu/ kriz/cifar.html

[4]https://grouplens.org/datasets/movielens/

RS task, we set a total of $T = 1000$ communication rounds and sample 20 clients per round with Adam optimizer Kingma and Ba (2014). In all datasets, each client trains for $E = 1$ epoch at the local update with a learning rate of $\eta_l = 0.01$. In our proposed FEDEVE , we set the global learning rate $\eta_g = 1$ for all experiments.

**Baselines.**  To evaluate the performance of FEDEVE , we compare it with several state-of-the-art FL methods: 1) The vanilla FL method FEDAVG McMahan et al. (2017), which is a widely used method for FL; 2) A client-side FL method FEDPROX Li et al. (2020b), which improves the model aggregation by adding a proximal term to the local update; 3) A server-side FL method FEDAVGM Hsu et al. (2019), which adapts the momentum in FL optimization; 4) A server-side FL method FEDOPT Reddi et al. (2020), which introduces adaptive optimization methods in FL. See more experimental details in the Appendix B.

### 4.2 ANALYSIS

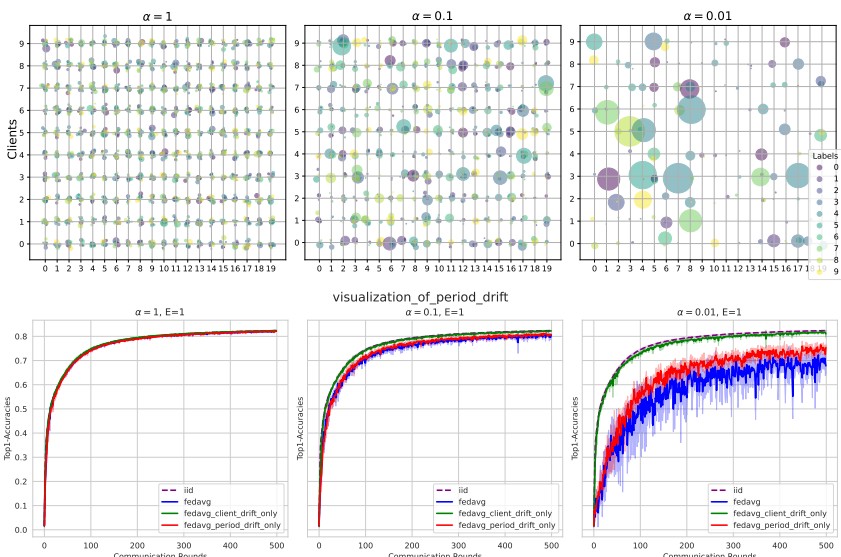

Figure 3: **Visualization of period drift and its impact on performance. (a) Visualization of period drift.** The color of the scatter points represents different classes, and the size denotes the number of samples of a given class on a particular client. When data is more non-iid (smaller $\alpha$), the heterogeneity of sampled data distributions becomes more pronounced both within a given communication round (*client drift*) and between different communication rounds (*period drift*). **(b) Visualization of its impact on performance.** It is revealed in cross-device FL when data is rather non-iid, period drift has a greater effect than client drift. Appendix B.2 for setting details.

**Visualizing the period drift and its impact.**  Figure 3 (a) visualizes the data distributions of these sampled clients. Client drift arises due to the shift in label distribution among sampled clients ***within a single round***, while period drift results from the shift in the data distribution of participating clients ***across different rounds***. The scatter points' size and distribution grow more diverse both within and across communication rounds as the value of $\alpha$ decreases (indicating increasing non-iid). The implications of these drifts on the global model's convergence are presented in Figure 3 (b). Utilizing the vanilla FEDAVG algorithm for illustration, we experimented with four settings: 1) FEDAVG with iid data; 2) FEDAVG experiencing only period drift; 3) FEDAVG subject to only client drift; and 4) FE-DAVG impacted by both drifts (See appendix B.2 for detailed settings). As heterogeneity intensifies, the effects of both drifts become evident. Specifically, in a highly non-iid environment ($\alpha = 0.01$), FEDAVG affected only by client drift yields results akin to the iid setting. In contrast, FEDAVG influenced solely by period drift significantly disrupts the stability and convergence of the FL process. The combination of both drifts results in the poorest performance, underlining that in cross-device FL, period drift poses a more considerable challenge to model convergence than client drift.

**The perforomance of FEDEVE .**  We evaluate our algorithm on real-world datasets and compare it with the relevant state-of-the-art methods in Tables 1 and 2. We conducted simulations on three datasets: FEMNIST, CIFAR-100, and MovieLens. The FEMNIST and Movielens datasets have a

Table 1: **Results on FEMNIST and CIFAR-100.** The best method is highlighted in **bold** fonts.

| Dataset | FEMNIST | | | | CIFAR-100 | | |
|---|---|---|---|---|---|---|---|
| Methods/NonIID | natural | $\alpha = 1$ | $\alpha = 0.1$ | $\alpha = 0.01$ | $\alpha = 1$ | $\alpha = 0.1$ | $\alpha = 0.01$ |
| FEDAVG | $82.37 \pm 0.18$ | $83.60 \pm 0.11$ | $82.02 \pm 0.23$ | $73.23 \pm 1.36$ | $47.04 \pm 0.21$ | $43.93 \pm 0.36$ | $30.11 \pm 0.53$ |
| FEDAVGM | $82.53 \pm 0.43$ | $83.67 \pm 0.10$ | $82.30 \pm 0.49$ | $74.96 \pm 2.34$ | $48.22 \pm 0.19$ | $44.74 \pm 0.40$ | $31.59 \pm 0.98$ |
| FEDPROX | $82.34 \pm 0.17$ | $83.58 \pm 0.11$ | $82.04 \pm 0.27$ | $74.16 \pm 1.19$ | $46.86 \pm 0.38$ | $43.74 \pm 0.27$ | $30.10 \pm 0.55$ |
| SCAFFOLD | $81.66 \pm 0.28$ | $83.06 \pm 0.14$ | $79.82 \pm 0.42$ | $5.13 \pm 0.00$ | $47.26 \pm 1.49$ | $36.36 \pm 4.98$ | $1.00 \pm 0.00$ |
| FEDOPT | $5.13 \pm 0.00$ | $81.86 \pm 0.38$ | $78.13 \pm 0.39$ | $5.13 \pm 0.00$ | $47.26 \pm 1.49$ | $45.43 \pm 1.18$ | $32.17 \pm 1.38$ |
| **FEDEVE** | $\mathbf{82.68 \pm 0.19}$ | $\mathbf{83.81 \pm 0.09}$ | $\mathbf{82.69 \pm 0.31}$ | $\mathbf{75.99 \pm 1.61}$ | $\mathbf{48.38 \pm 0.24}$ | $\mathbf{45.68 \pm 0.16}$ | $\mathbf{32.68 \pm 0.62}$ |

Table 2: **Results on MovieLens-1M.** The best method is highlighted in **bold** fonts.

| | AUC | HR@5 | HR@10 | NGCG@5 | NGCG@10 |
|---|---|---|---|---|---|
| FEDAVG | $0.7633 \pm 0.0065$ | $0.2774 \pm 0.0100$ | $0.4294 \pm 0.0120$ | $0.1835 \pm 0.0058$ | $0.2324 \pm 0.0064$ |
| FEDAVGM | $0.7555 \pm 0.0128$ | $0.2705 \pm 0.0384$ | $0.4290 \pm 0.0196$ | $0.1771 \pm 0.0319$ | $0.2280 \pm 0.0257$ |
| FEDPROX | $0.7819 \pm 0.0033$ | $0.2700 \pm 0.0129$ | $0.4279 \pm 0.0083$ | $0.1803 \pm 0.0078$ | $0.2310 \pm 0.0065$ |
| FEDOPT | $0.7751 \pm 0.0085$ | $0.2868 \pm 0.0055$ | $0.4392 \pm 0.0101$ | $0.1886 \pm 0.0044$ | $0.2377 \pm 0.0040$ |
| FEDEVE | $\mathbf{0.7967 \pm 0.0016}$ | $\mathbf{0.2916 \pm 0.0077}$ | $\mathbf{0.4460 \pm 0.0088}$ | $\mathbf{0.1924 \pm 0.0039}$ | $\mathbf{0.2407 \pm 0.0037}$ |

naturally-arising client partitioning setting in real-world FL scenarios, making them highly representative. For FEMNIST and CIFAR-100 datasets, each of the datasets includes three non-iid settings, established through the Dirichlet distribution partition method (Hsu et al., 2019). *Generally, the results show that our proposed algorithm,* FEDEVE *, consistently outperforms the baselines, and the performance gains are more dominant in more non-iid settings ($\alpha = 0.01$).* We also conduct experiments with different local epochs ($E$), please refer to the appendix B.3 for the setting details. Also, our FEDEVE has more leading advantages in RS experiments, indicating its large potential in real-world industrial applications. Our method can better utilize the server-side adaptation through the Bayesian filter's predict-observer framework. Besides, it is important to note that our method does not introduce other hyperparameters while these baselines have multiple hyperparameters to tune, which means that our FEDEVE is more flexible and advantageous in real-world practices.

**Analysis of Kalman Gain in FEDEVE .** We conducted an in-depth analysis of the Kalman Gain $K$ of FedEve under various experimental settings, incorporating four levels of data heterogeneity and various local epochs, as shown in Figure 4. We observed that as data heterogeneity increases (i.e., as the value of $\alpha$ decreases), the Kalman Gain $K$ progressively enlarges. With the rise of data heterogeneity, the period drift starts to play a more dominant role. In this context, the primary role of Kalman

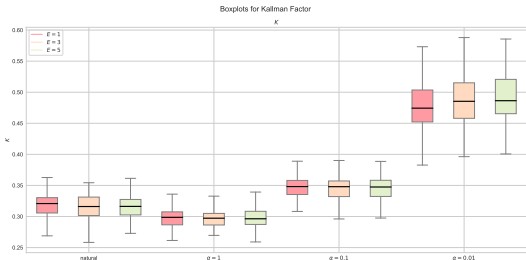

Figure 4: **Boxplots for Kalman Factors**

Gain $K$ is to adjust the weights between global and local updates, as depicted by Equation (12b) and (12c). Further, according to Equation (12d), the model update tends to trust local updates more, stabilizing the optimization process. For varied counts of local updates, the relative change in Kalman Gain $K$ is marginal. This is primarily because, in cross-device FL, the client drift is not a pivotal or dominant factor, which aligns with our prior analysis in Figure 3.

## 5 CONCLUSION

In this work, we explored the impact of client drift and period drift on the performance of cross-device FL, discovering that period drift can be particularly harmful as data heterogeneity increases. To solve this challenge, we introduced a novel predict-observe framework and a method, FEDEVE , that views these drifts as noise associated with prediction and observation. By integrating these two sources in a principled way, we provided a better estimation of model update steps, reducing variance and improving the stability and convergence speed of FL. Our theoretical and empirical evaluations demonstrated that FEDEVE significantly outperforms alternative methods, shedding light on future directions for improving efficiency in FL.

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

# A APPENDIX

## READING GUIDE

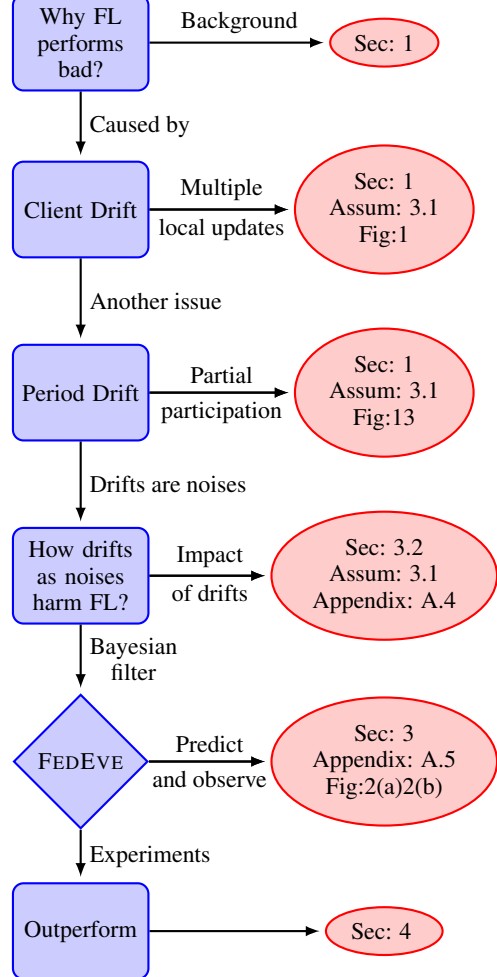

In this paper, we discuss the issue of why Federated Learning (FL) sometimes performs poorly, discussed in Section 1. We discuss the specific issues of "Client Drift" and "Period Drift", which are two of the many potential reasons for this bad performance. The problem of Client Drift is introduced in Section 1, with the assumptions made in this respect laid out in assumption 3.1. This section will explore how multiple local updates contribute to this drift and the overall performance degradation of FL. Subsequently, we shift our focus to Period Drift, which also contributes to FL's challenges. This issue is further examined in Section 1 and is guided by the assumptions set forth in assumption 3.1. In particular, we study the impact of partial client participation on this type of drift. These drifts can be considered as noises into the system, we explore the detrimental effects of this noise on FL in Section 3.2. The supplementary assumptions and related analysis of noise are available in Appendix A.4. To tackle these challenges, we propose a Bayesian filtering approach called 'FEDEVE' in Section 3. This novel method, described in more detail in Appendix A.5, utilizes a fusion of prediction and observation to mitigate the drifts. Finally, in Section 4, we present experimental results demonstrating the effectiveness of our proposed method in enhancing FL's performance, and thereby outperforming other existing solutions. Details about the experiment and the obtained results can be found in this section. In summary, this paper delves deep into the problems plaguing FL and proposes a viable solution to improve its performance. We hope this provides a useful roadmap for the reader.

## A.1 THE ANALOGY BETWEEN *Drift* IN FL AND THE *Noise* OF CENTRALIZED SGD

Federated learning possesses unique characteristics compared to traditional centralized optimization, such as client sampling, multiple local epochs, and non-iid data distribution. In this context, drifts in federated learning can be viewed as noises to the training dynamics. More specifically, period drift, originating from non-iid data and partial participation (only a subset of clients participate in each round), can be likened to the implementation of a mini-batch technique in the full-batch gradient descent of centralized training (Ziyin et al., 2021). Here, the distinction is that in centralized optimization issues, each batch is iid, and each batch's data distribution closely mirrors the overall distribution, albeit with a noise component. This noise becomes remarkably pronounced in federated learning, given that client data is non-iid. Client drift, arising from non-iid data and multiple local updates (where each client runs local SGD with multiple steps), is a well-structured noise (Lin et al., 2018). Due to the combined impact of client drift and period drift, the situation can be perceived as adding a noise term to the original model (or gradient). The research outlined in this paper based on the analogy between the drifts in federated learning and the noises in centralized SGD.

## A.2 THE JUSTIFICATION OF GAUSSIAN-LIKE NOISE ASSUMPTION 3.1

**Assumption A.1** (3.1). *The aggregated model parameters on the server $w_{server}$, can be represented as the sum of the optimal parameters $w^*$ and a drift (noise) that follows a normal distribution $w_{drift} \sim \mathcal{N}(0, \sigma_{drift}^2)$:*

$$w_{server} = w^* + w_{drift} \leftarrow noise, \tag{14}$$

where $w^*$ represents the optimal parameters obtained through the use of stochastic gradient descent (SGD), $w_{drift}$ represents the noise term caused by factors such as client sampling, multiple local epochs, and non-iid data distribution that we assume a normal distribution, and $w_{server}$ represents the aggregated model parameters also follows a normal distribution $w_{server} \sim \mathcal{N}(w^*, \sigma_{drift}^2)$, with the expectation of the aggregate model parameters being equal to the optimal parameters, i.e. $\mathbb{E}[w_{server}] = w^*$.

In this paper, we conceptualize the aggregated model parameters on the server as the summation of optimal parameters and a certain drift (or noise), represented as: $w_{server} = w^* + w_{drift}$. We also assume that $w_{drift}$ is subject to a normal (Gaussian-like) distribution, and justify this assumption by demonstrating its prevalence, and explaining it in FL.

From a historical standpoint, modeling noise in dynamic systems as a Gaussian-like distribution is a widely accepted practice. This dates back to (Kramers, 1940), and many studies analyzing Stochastic Gradient Descent (SGD) optimization have emphasized the Gaussian nature of noise on gradients or parameters (Mandt et al., 2017; Zhu et al., 2019; Simsekli et al., 2019; Ziyin et al., 2021). This assumption of Gaussianity for SGD noise is justified by Wu et al. (2020), which guarantees the SGD noise's convergence to a specific infinite divisible distribution. This falls under the Gaussian class provided the noise's second moment is finite (as per Lindeberg's condition). While it has been proposed that the noise in SGD might be better represented by $S\alpha S$ noise (Simsekli et al., 2019), this idea has been challenged and redirected back to the earlier proposed Gaussian noise model (Xie et al., 2021; Battash and Lindenbaum, 2023).

We further elucidate the occurrence of Gaussian noise in the context of FL. The Lindeberg-Feller Central Limit Theorem (CLT) (Lindeberg, 1922) plays a key role in explaining the prevalence of the normal distribution. It posits that the sum (or average) of random variables gravitates towards a normal distribution (no need to assume iid of these random variables themselves), irrespective of the individual distributions of these variables. In FL, the emergence of noise can be attributed to partial participation (referred to as period drift) and multiple local updates (referred to as client drift). The drifted model that we observe, $w_{server}$, is typically the result of the combination of these factors, making the normal distribution an apt model for characterizing the noise.

**The impact of noise on generalization** In order to investigate the effect of the deviation on performance in FL, we utilize a regression optimization objective as in previous studies, such as (Zhang et al., 2019) and (Wu et al., 2018):

$$\hat{\mathcal{L}}(w) = \frac{1}{2}(w + w_{drift})^T A(w + w_{drift}),$$

where $w_{drift} \sim \mathcal{N}(0, \sigma^2)$ is the drift caused by the characteristics of FL. Therefore, the generalization error can be formulated as:

$$\mathcal{L}\left(w^t\right) = \mathbb{E}\left[\hat{\mathcal{L}}\left(w^t\right)\right] = \frac{1}{2}\mathbb{E}\left[\sum_i a_i \left(w_i^{t^2} + \sigma_i^2\right)\right]$$
$$= \frac{1}{2}\sum_i a_i \left(\mathbb{E}\left[w_i^t\right]^2 + \mathbb{V}\left[w_i^t\right] + \sigma_i^2\right),$$

where $A$ is the matrix of quadratic form of the MSE loss function, and $a_i$ is the elements of $A$. As results, the generalization error can also be decomposed into three components: bias, variance, and noise. The noise component in FL context is further influenced by factors such as client sampling, multiple local epochs, and non-iid data distribution, leading to a much larger overall generalization error compared to centralized SGD. This formulation reveals the reason of why FL usually performs worse than centralized training. Thus, our goal is to reduce the variance of drift $\sigma^2$ in order to improve both the convergence and performance of the model.

### A.3 THE JUSTIFICATION OF INDEPENDENCE ASSUMPTION 3.2 OF CLIENT DRIFT AND PERIOD DRIFT

**Assumption A.2** (3.2). *The initialization model parameters are independent of all period drifts $Q_t$ and client drifts $R_t$ at each communication round, that is $w_0 \perp Q_0, Q_1, \cdots, Q_t$ and $w_0 \perp R_0, R_1, \cdots, R_t$.*

This assumption can be justified from various perspectives, demonstrating its reasonableness:

- **Independence of the initial model parameters from other noise variables:** This assumption suggests that there is no direct relationship between the initial model parameters ($w_0$) and period drifts or client drifts. This is a reasonable assumption since initial model parameters are typically determined prior to training and hence are not influenced by any noise processes.
- **Independence of client drifts between each communication round:** According to this assumption, client drifts ($R_0, R_1, \cdots, R_t$) in different communication rounds are independent of each other. The client drift is influenced by data heterogeneity and multiple local updates. A higher degree of data heterogeneity and an increased number of local updates can result in greater client drift. Each client has its own fixed client drift(Guo et al., 2021), and the client drift in each communication round doesn't impact other rounds, the assumption of client drift independence is reasonable.
- **Independence of period drifts between each communication round:** This assumption contends that the period drifts ($Q_0, Q_1, \cdots, Q_t$) across different communication rounds are independent. Period drift is influenced by data heterogeneity and client sampling. Though period drift may be caused by biased client sampling due to factors like time and geographic locations, leading to a dependence between period drifts in different communication rounds, this paper considers the case where random client sampling occurs in each communication round. Here, one round of client sampling doesn't affect others, thus rendering the independence of period drift reasonable.
- **Independence between period drifts and client drifts at each communication round:** This assumption argues that the client drifts ($R_0, R_1, \cdots, R_t$) and the period drifts ($Q_0, Q_1, \cdots, Q_t$) at each communication round are independent. While both client drift and period drift are influenced by data heterogeneity, they are conditionally independent given the heterogeneous data. We offer a causal graph to depict their relationships:

multiply local updates $\longrightarrow$ client drift $\longleftarrow$ data heterogeneity $\longrightarrow$ period drift $\longleftarrow$ client sampling

This graph indicates that data heterogeneity is a common cause of client drift and period drift, and varying levels of data heterogeneity result in different magnitudes of client drift and period drift. However, given that the heterogeneous data is constant across clients (i.e., given D), we can express P(client drift, period drift—D) = P(client drift—D) * P(period drift—D). Indeed, conditioning on a given heterogeneous data set is a fundamental assumption in Federated Learning.

### A.4 THE INDEPENDENCE OF NOISE

**Lemma A.3** (3.3). *(Independence of Noise). the noise present in the prediction and observation at each communication round is independent of the current state of the model, specifically, $w_t \perp Q_t$ and $w_t \perp R_t$.*

To prove this, we will first need to understand the relationship between the variables $w_{t+1}$, $Q_t$, and $R_t$. In the given context, $w_{t+1}$ represents the state of the model at a particular communication round (say, round $t+1$). It is affected by the values of the period drift ($Q_t$) and client drift ($R_t$) at the previous round. This is represented by the state transfer function $w_{t+1} = T_t(w_t, Q_t, R_t)$. However, this relationship does not imply that $w_{t+1}$ is dependent on $Q_t$ or $R_t$. To see why, let's look at how $w_t$ is formed. Using the state transfer function, we can express $w_t$ as:

$$
\begin{aligned}
w_t &= T_{t-1}(w_{t-1}, Q_{t-1}, R_{t-1}), \\
w_{t-1} &= T_{t-2}(w_{t-2}, Q_{t-2}, R_{t-2}), \\
&\vdots \\
w_2 &= T_1(w_1, Q_1, R_1), \\
w_1 &= T_0(w_0, Q_0, R_0),
\end{aligned}
\tag{15}
$$

From this chain of equations, it is evident that $w_t$ is not only a function of the current round's period drift $Q_t$ and client drift $R_t$, but also of their past values and the past values of $w_t$ itself. In a more generalized form, we can write this as: $w_t = T(w_0, Q_0, Q_1, \cdots, Q_{t-1}, R_0, R_1, \cdots, R_{t-1})$. Assumption 3.2 states that the period drift and client drift are independent of each other at each communication round and also independent of the initial model parameters. That is, $w_0 \perp Q_0 \perp Q_1 \perp \cdots Q_t \perp R_0 \perp R_1 \perp \cdots R_t$. This assumption implies that the previous states of $w_t$ ($w_{t-1}$, $w_{t-2}$, and so on) do not have any influence on the current values of $Q_t$ and $R_t$. In other words, the noise present at each round is independent of the model's current state. Thus, $w_t$ is independent of $Q_t$ and $R_t$, denoted as $w_t \perp Q_t \perp R_t$. Therefore, it can be concluded that the noise present in the prediction and observation at each communication round is indeed independent of the current state of the model, thereby confirming the independence of noise. This statement about the independence of noise is significant because it confirms that the noise encountered during each communication round does not affect the model's state. This means that the model is robust and not affected by random perturbations, which is a desirable property in any machine learning model.

## A.5 MODEL UPDATE WITH BAYESIAN FILTER

### A.5.1 BAYESIAN FILTER

This section begins by describing the main idea behind the approach: the combination of prediction and observation models using a Bayesian filter, which is a statistical tool for estimating an unknown probability density function (PDF) based on observations. The prediction process is explained using the concept of cumulative distribution functions (CDFs), which are functions giving the probability that a random variable is less than or equal to a certain value. The prediction process is characterized by the distribution:

$$
\begin{aligned}
F_{\hat{w}_{t+1}}^{-}(w) &= P\left(\hat{w}_{t+1} \leq w\right) \\
&= \sum_{u=-\infty}^{w} P\left(\hat{w}_{t+1} = u\right) \\
&= \sum_{u=-\infty}^{w} \sum_{v=-\infty}^{+\infty} P\left(\hat{w}_{t+1} = u \mid w_t = v\right) P\left(w_t = v\right) \\
&= \sum_{u=-\infty}^{w} \sum_{v=-\infty}^{+\infty} P\left[\hat{w}_{t+1} - f\left(w_t\right) = u - f(v) \mid w_t = v\right] P\left(w_t = v\right) \\
&= \sum_{u=-\infty}^{w} \sum_{v=-\infty}^{+\infty} P\left[Q_t = u - f(v) \mid w_t = v\right] P\left(w_t = v\right) \quad \Rightarrow \text{Prediction Equation} \\
&= \sum_{u=-\infty}^{w} \sum_{v=-\infty}^{+\infty} P\left[Q_t = u - f(v)\right] P\left(w_t = v\right) \quad \Rightarrow \text{Lemma (3.3)} \\
&= \sum_{u=-\infty}^{w} \left\{ \lim_{\epsilon \to 0} \sum_{v=-\infty}^{+\infty} f_{Q_t}[u - f(v)] \cdot \epsilon \cdot f_{w_t}^{+}(v) \cdot \epsilon \right\} \\
&= \sum_{u=-\infty}^{w} \left\{ \lim_{\epsilon \to 0} \int_{-\infty}^{+\infty} f_{Q_t}[u - f(v)] f_{w_t}^{+}(v) \mathrm{d}v \cdot \epsilon \right\} \\
&= \int_{-\infty}^{w} \int_{-\infty}^{+\infty} f_{Q_t}[u - f(v)] f_{w_t}^{+}(v) \mathrm{d}v \, \mathrm{d}u \\
&= \int_{-\infty}^{w} \int_{-\infty}^{+\infty} f_{Q_t}[w - f(v)] f_{w_t}^{+}(v) \mathrm{d}v \, \mathrm{d}w
\end{aligned}
\tag{16}
$$

The first three steps apply the definition of the cumulative distribution function (CDF), which is the sum of probabilities up to a certain point. In the fourth step, the change of variables is used to switch from $u$ to $v$. The fifth and sixth steps apply Lemma 3.3, which states that the drift is independent of the weights $w_t$. The last three steps show how to convert the sum to an integral, which is a common method in probability theory for dealing with continuous random variables. Finally, the PDF of the prediction is obtained by taking the derivative of the CDF, which can be expressed as:

$$f_{\hat{w}_{t+1}}^{-}(w) = \frac{\mathrm{d}F_{\hat{w}_{t+1}}^{-}(w)}{\mathrm{d}w} = \int_{-\infty}^{+\infty} f_{Q_t}[w - f(v)]f_{w_t}^{+}(v)\mathrm{d}v \tag{17}$$

The observation process is also characterized by a PDF. Similar steps are used to manipulate and simplify the expression for the PDF of the observed value $w_{t+1}$, given the predicted value $\hat{w}_{t+1}$. Specifically:

$$\begin{aligned}
f_{\tilde{w}_{t+1}|\hat{w}_{t+1}}(w_{t+1} \mid w) &= \lim_{\epsilon \to 0} \frac{F_{\tilde{w}_{t+1}|\hat{w}_{t+1}}(w_{t+1} + \epsilon \mid w) - F_{\tilde{w}_{t+1}|\hat{w}_{t+1}}(w_{t+1} \mid w)}{\epsilon} \\
&= \lim_{\epsilon \to 0} \frac{P(w_{t+1} \leq \tilde{w}_{t+1} \leq w_{t+1} + \epsilon \mid \hat{w}_{t+1} = w)}{\epsilon} \\
&= \lim_{\epsilon \to 0} \frac{P[w_{t+1} - h(w) \leq \tilde{w}_{t+1} - h(\hat{w}_{t+1}) \leq w_{t+1} - h(w) + \epsilon \mid \hat{w}_{t+1} = w]}{\epsilon} \\
&= \lim_{\epsilon \to 0} \frac{P[w_{t+1} - h(w) \leq R_t \leq w_{t+1} - h(w) + \epsilon \mid \hat{w}_{t+1} = w]}{\epsilon} \\
&= \lim_{\epsilon \to 0} \frac{P[w_{t+1} - h(w) \leq R_t \leq w_{t+1} - h(w) + \epsilon]}{\epsilon} \Rightarrow \text{Lemma (3.3)} \\
&= \lim_{\epsilon \to 0} \frac{F_{R_t}[w_{t+1} - h(w) + \epsilon] - F_{R_t}[w_{t+1} - h(w)]}{\epsilon} \\
&= f_{R_t}[w_{t+1} - h(w)].
\end{aligned} \tag{18}$$

The first two steps apply the definition of the probability density function (PDF), which is the derivative of the cumulative distribution function (CDF). The third and fourth steps use a change of variables to express the PDF in terms of the difference between the observed and predicted values. The fifth step applies the independence Lemma (3.3) to simplify the conditional probability. The sixth step calculates the limit to arrive at the PDF of the observation process. As a consequence of combining the prediction and observation distributions, a fused model can be obtained, which can be described by its own probability density function (PDF) as:

$$f_{w_{t+1}}^{+}(w) = \eta_t \cdot f_{\tilde{w}_{t+1}|\hat{w}_{t+1}}(\tilde{w}_{t+1} \mid \hat{w}_{t+1}) \cdot f_{\hat{w}_{t+1}}^{-}(w) = \eta_t \cdot f_{R_t}[\tilde{w}_{t+1} - h(\hat{w}_{t+1})] \cdot f_{\hat{w}_{t+1}}^{-}(w), \tag{19}$$

where

$$\eta_t = \left[\int_{-\infty}^{+\infty} f_{\tilde{w}_{t+1}|\hat{w}_{t+1}}(\tilde{w}_{t+1} \mid \hat{w}_{t+1}) f_{\hat{w}_{t+1}}^{-}(w)\mathrm{d}w\right]^{-1} = \left\{\int_{-\infty}^{+\infty} f_{R_t}[\tilde{w}_{t+1} - h(\hat{w}_{t+1})] f_{\hat{w}_{t+1}}^{-}(w)\mathrm{d}w\right\}^{-1}. \tag{20}$$

The PDF of the fused model is obtained by multiplying the PDFs of the prediction and observation processes, normalized by a factor $\eta_t$. The process of updating the fused model, also known as the Bayesian filter, can be summarized in the following steps:

$$\begin{aligned}
f_{w_t}^{+}(w) &\overset{\text{predict}}{\Longrightarrow} f_{\hat{w}_{t+1}}^{-}(w) = \int_{-\infty}^{+\infty} f_{Q_t}[w - f(v)]f_{w_t}^{+}(v)\mathrm{d}v \\
&\overset{\text{observe}}{\Longrightarrow} f_{w_{t+1}}^{+}(w) = \eta_t \cdot f_{R_t}[w_{t+1} - h(w)] \cdot f_{\hat{w}_{t+1}}^{-}(w),
\end{aligned} \tag{21}$$

where $\eta_t = \left\{\int_{-\infty}^{+\infty} f_{R_t}[\tilde{w}_{t+1} - h(\hat{w}_{t+1})] f_{\hat{w}_{t+1}}^{-}(w)\mathrm{d}w\right\}^{-1}$. The fused model combines the prediction and observation distributions, and it describes the sequential steps of the Bayesian filter: starting with the PDF at time $t$, a prediction is made for the PDF at time $t + 1$, and then this prediction is updated based on the observation to obtain the PDF at time $t + 1$. The final estimation of the parameter can be obtained as a result of these update steps and can be represented as:

$$\hat{w}_{t+1} = E\left[f_{w_{k+1}}^{+}(w)\right] = \int_{-\infty}^{+\infty} w f_{w_{k+1}}^{+}(w)\mathrm{d}w, \tag{22}$$

which is done by calculating the expected value of the PDF at time $t + 1$. This is performed by multiplying the parameter $w$ by its probability density and integrating over all possible values of $w$. The integral provides a single, average value for $w$ weighted by its probability density, which serves as the final estimate of the parameter.

### A.5.2 FEDEVE

In this section, we delve into the derivation of the FEDEVE algorithm using the Bayesian filter. The model update process within the FEDEVE algorithm is outlined as follows: Firstly, we calculate the predictive value of the model's parameters ($\hat{w}_{t+1}$) at the next time step using the current parameters ($w_t$) and the step-size scaled momentum ($\eta_g M_t$):

$$\hat{w}_{t+1} = w_t - \eta_g M_t, \tag{23}$$

The inverse of the variance at time $t + 1$ ($\hat{\sigma}_{t+1}^2$) is determined as the sum of the predicted variance at time $t$ ($\sigma_t$) and the squared variance associated with the process noise ($\sigma_{Q_t}^2$):

$$\hat{\sigma}_{t+1}^2 = \sigma_t + \sigma_{Q_t}^2, \tag{24}$$

The Kalman Gain ($K$) is computed as the ratio of the inverse of the variance at time $t+1$ to the sum of the inverse of the variance at time $t + 1$ and the variance of the observation noise ($\sigma_{R_t}^2$):

$$K = \frac{\hat{\sigma}_{t+1}^2}{\hat{\sigma}_{t+1}^2 + \sigma_{R_t}^2}, \tag{25}$$

The momentum for the next time step ($M_{t+1}$) is obtained by adjusting the current momentum ($M_t$) based on the difference between the observed value ($\Delta \tilde{w}_t$) and the current momentum:

$$M_{t+1} = M_t + K(\Delta \tilde{w}_t - M_t), \tag{26}$$

The parameters ($w_{t+1}$) for the next time step are calculated by subtracting the step-size scaled updated momentum from the current parameters:

$$w_{t+1} = w_t - \eta_g M_{t+1}, \tag{27}$$

Finally, the predicted variance for the next time step ($\sigma_t^2$) is computed by scaling the inverse of the variance at time $t + 1$ by $(1 - K)$:

$$\sigma_{t+1}^2 = (1 - K)\hat{\sigma}_{t+1}^2. \tag{28}$$

A key assumption for this derivation is that the two random variables, $A$ and $B$, follow a normal distribution. Specifically, $A$ is assumed to be distributed as $\mathcal{N}\left(\mu_A, \sigma_A^2\right)$, and $B$ is assumed to be distributed as $\mathcal{N}\left(\mu_B, \sigma_B^2\right)$. Given these assumptions, it can be mathematically proven that the sum and the product of $A$ and $B$ also follow a normal distribution. In particular, we have:

$$A + B \sim \mathcal{N}\left(\mu_A + \mu_B, \sigma_A^2 + \sigma_B^2\right), \tag{29}$$

$$A \times B \sim \mathcal{N}\left(\frac{\mu_A \sigma_B^2 + \mu_B \sigma_A^2}{\sigma_A^2 + \sigma_B^2}, \frac{\sigma_A^2 \sigma_B^2}{\sigma_A^2 + \sigma_B^2}\right), \tag{30}$$

In the context of the predict-observe framework and the Bayesian filter, we make a few key assumptions: firstly, $w_t$ is distributed as $\mathcal{N}\left(w_t, \sigma_t^2\right)$; secondly, $Q_t$ is distributed as $\mathcal{N}\left(0, \sigma_{Q_t}^2\right)$; and finally, the momentum $M_t$ is considered a non-random variable. Specializing the prediction function as a linear function, as shown in Equation (10), leads us to the following result:

$$\hat{w}_{t+1} \sim \mathcal{N}\left(w_t - \eta_g M_t, \sigma_t^2 + \sigma_{Q_t}^2\right), \tag{31}$$

This is essentially equivalent to the equations (12a) and (12b) in the model update process. Moreover, we set $\sigma_{t+1}^{-2} = \sigma_t^2 + \sigma_{Q_t}^2$. The distribution of $w_{t+1}$ is considered as the posterior, which is calculated by applying the Bayesian formula and combining the product of the likelihood and the prior. Here, the likelihood corresponds to the observation, and the prior corresponds to the prediction. The observed $\tilde{w}_{t+1}$ is distributed as follows:

$$\tilde{w}_{t+1} \sim \mathcal{N}\left(\hat{w}_{t+1} - \eta_g \Delta \tilde{w}_t, \sigma_{R_t}^2\right). \tag{32}$$

To calculate the product of the prior and the likelihood, we evaluate the proportionality coefficient $K = \frac{\hat{\sigma}_{t+1}^2}{\hat{\sigma}_{t+1}^2 + \sigma_{R_t}^2}$. We can then assert that $w_{t+1}$ also adheres to a normal distribution:

$$\tilde{w}_{t+1} \sim \mathcal{N}\left(w_t - \eta_g((1 - K)M_t + K\Delta \tilde{w}_t), (1 - K)\hat{\sigma}_{t+1}^2\right), \tag{33}$$

This result serves as a stepping stone for the subsequent round of computation. Consequently, the variance of $w_{t+1}$ is minimized as:

$$\sigma_{\text{fused}}^2 = \frac{\hat{\sigma}_{t+1}^2 \sigma_{R_t}^2}{\hat{\sigma}_{t+1}^2 + \sigma_{R_t}^2}. \tag{34}$$

In summary, the above proof demonstrates how the Bayesian filter can be used to derive the model update process of the FEDEVE algorithm. The predict-observe framework and the feature of normal distribution are key elements in this derivation.

## A.6 PSEUDO-CODE

The pseudo-code of FEDEVE is depicted in Algorithm 1.

---

**Algorithm 1 FEDEVE** The selected clients are indexed by $k$; $E$ is the number of local epochs, and $\eta_l$ is the local learning rate.

---

**Server executes:**
   initialize $w_0$
   **for** each round $t = 1, 2, \ldots, T$ **do**
      $\hat{w}_{t+1} \leftarrow w_t - \eta_g M_t$ as in Equation (12) in the main paper // predict
      $\mathcal{S}_t \leftarrow$ randomly select $|\mathcal{S}_t|$ clients
      **for** each client $k \in \mathcal{S}_t$ **in parallel do**
         $w_t^k \leftarrow$ ClientUpdate$(k, \hat{w}_{t+1})$
      **end for**
      $\Delta \tilde{w}_t = \sum_{k \in \mathcal{S}_t} p_k \left( \hat{w}_{t+1} - w_t^k \right)$ // observe
      *Model update:* executes Equations (13)-(17) in the main paper
   **end for**

**ClientUpdate**$(k, w)$:   // run on client $k$
   $\mathcal{B} \leftarrow$ (split local data into batches of size)
   **for** each local epoch $i$ from 1 to $E$ **do**
      **for** batch $b \in \mathcal{B}$ **do**
         $w \leftarrow w - \eta_l F_k(w; b)$
      **end for**
   **end for**
   return $w$ to server

---

## B    EXPERIMENTAL DETAILS

**Implementation.**    All the experiments are implemented using PyTorch, a popular machine learning library. We simulate the federated learning environment, including clients, and run all experiments on a deep learning server equipped with an NVIDIA GTX 2080 ti GPU.

### B.1    EVALUATION METRIC.

For the RS task, the model performance is evaluated using the following metrics: area under curve (AUC), Hit Ratio (HR) and Normalized Discounted Cumulative Gain (NDCG). For the CV task, the model performance is measured by the widely used Top-1 accuracy metric. In the experiments, for the CTR (Click-Through Rate) task, the model performance is evaluated using the following metrics: area under curve (AUC), Hit Ratio (HR) and Normalized Discounted Cumulative Gain (NDCG).

$$\text{AUC} = \frac{\sum_{x_0 \in D_T} \sum_{x_1 \in D_F} \mathbf{1}\left[f\left(x_1\right) < f\left(x_0\right)\right]}{|D_T| \, |D_F|},$$

$$\text{HitRate@K} = \frac{1}{|\mathcal{U}|} \sum_{u \in \mathcal{U}} \mathbf{1}\left(R_{u,g_u} \leq K\right),$$

$$\text{NDCG@K} = \sum_{u \in \mathcal{U}} \frac{1}{|\mathcal{U}|} \frac{2^{\mathbf{1}(R_{u,g_u} \leq K)} - 1}{\log_2\left(\mathbf{1}\left(R_{u,g_u} \leq K\right) + 1\right)},$$

where $\mathcal{U}$ is the set of users, $\mathbf{1}$ is the indicator function, $R_{u,g_u}$ is the rank generated by the model for the ground truth item $g_u$, $f$ is the model being evaluated, and $D_T$ and $D_F$ are the positive and negative sample sets in the testing data, respectively. For the image classification task, the model performance is measured by the widely used Top-1 accuracy metric.

### B.2    DETAILED SETTINGS OF FIGURE 3 IN THE MAIN PAPER.

- **Figure 3 (a).** To provide a clearer illustration, we displayed the label distribution of the 10-digit classes, rather than the complete 62 classes in the original FEMNIST dataset, for 20 communication rounds.

- **Figure 3 (b).** 1) FEDAVG *with iid data with no period or client drift:* FEDAVG with iid data, where training data is randomly partitioned among all clients, resulting in no period or client drift; 2) FEDAVG *with only period drift:* non-iid data is initially partitioned, but the training data of the sampled clients is randomly reshuffled and iid-distributed evenly among clients in each round, resulting in period drift but no client drift; 3) FEDAVG *with only client drift:* iid data is initially partitioned, but the training data of the sampled clients in each round is re-partitioned in non-iid setting, resulting in client drift but no period drift; 4) FEDAVG *with both period and client drift:* FEDAVG with non-iid data, where data is partitioned in non-iid setting, resulting in both period and client drift.

To make the results more convincing, we conducted more experiments on FEMNIST. Specifically, we add experiments with various local epochs and data heterogeneity. Each experiment is repeated 5 times, and the results are shown as follow:

### B.3    PERFORMACE OF FEDEVE ON FEMNIST WITH DIFFERENT LOCAL EPOCHS

Table 3: **Results on FEMNIST with different** $\alpha$ **and** $E$**.** The best method is highlighted in **bold** fonts.

| Method | Natural | | | $\alpha = 1$ | | | $\alpha = 0.1$ | | | $\alpha = 0.01$ | | |
|---|---|---|---|---|---|---|---|---|---|---|---|---|
| | $E = 1$ | $E = 3$ | $E = 5$ | $E = 1$ | $E = 3$ | $E = 5$ | $E = 1$ | $E = 3$ | $E = 5$ | $E = 1$ | $E = 3$ | $E = 5$ |
| FEDAVG | $82.46 \pm 0.18$ | $81.95 \pm 0.26$ | $66.38 \pm 30.63$ | $83.64 \pm 0.11$ | $83.57 \pm 0.12$ | $83.38 \pm 0.09$ | $82.12 \pm 0.23$ | $81.91 \pm 0.23$ | $81.70 \pm 0.26$ | $74.51 \pm 1.36$ | $73.43 \pm 1.73$ | $72.67 \pm 1.39$ |
| FEDAVGM | $82.55 \pm 0.43$ | $81.99 \pm 0.42$ | $50.97 \pm 37.43$ | $83.67 \pm 0.10$ | $83.79 \pm 0.11$ | $83.65 \pm 0.08$ | $82.36 \pm 0.49$ | $82.23 \pm 0.26$ | $82.16 \pm 0.34$ | $75.18 \pm 2.34$ | $74.20 \pm 2.81$ | $73.79 \pm 3.13$ |
| FEDPROX | $82.43 \pm 0.17$ | $81.90 \pm 0.27$ | $51.07 \pm 37.51$ | $83.62 \pm 0.11$ | $83.52 \pm 0.14$ | $67.65 \pm 31.26$ | $82.17 \pm 0.27$ | $82.12 \pm 0.19$ | $81.94 \pm 0.30$ | $75.07 \pm 1.19$ | $74.38 \pm 1.46$ | $60.22 \pm 27.56$ |
| SCAFFOLD | $81.66 \pm 0.28$ | $81.08 \pm 0.36$ | $80.76 \pm 0.30$ | $83.18 \pm 0.14$ | $82.75 \pm 0.16$ | $82.46 \pm 0.16$ | $79.82 \pm 0.42$ | $79.00 \pm 0.65$ | $78.44 \pm 0.70$ | $5.13 \pm 0.00$ | $5.13 \pm 0.00$ | $5.13 \pm 0.00$ |
| FEDOPT | $5.13 \pm 0.00$ | $5.13 \pm 0.00$ | $5.13 \pm 0.00$ | $81.86 \pm 0.38$ | $35.90 \pm 37.69$ | $35.66 \pm 37.39$ | $78.13 \pm 0.39$ | $5.13 \pm 0.00$ | $5.13 \pm 0.00$ | $5.13 \pm 0.00$ | $5.13 \pm 0.00$ | $5.13 \pm 0.00$ |
| FEDEVE | $\mathbf{82.66 \pm 0.19}$ | $\mathbf{82.20 \pm 0.20}$ | $\mathbf{81.93 \pm 0.16}$ | $\mathbf{83.81 \pm 0.09}$ | $\mathbf{83.88 \pm 0.08}$ | $\mathbf{83.72 \pm 0.05}$ | $\mathbf{82.69 \pm 0.31}$ | $\mathbf{82.66 \pm 0.18}$ | $\mathbf{82.52 \pm 0.19}$ | $\mathbf{75.99 \pm 1.61}$ | $\mathbf{75.00 \pm 2.24}$ | $\mathbf{74.56 \pm 2.05}$ |

The table showcases the results on the FEMNIST dataset across various methods, specifically: FEDAVG , FEDAVGM , FEDPROX , SCAFFOLD , FEDOPT , and FEDEVE , with different settings of parameters $\alpha$ and $E$. FEDEVE method stands out consistently as the superior approach across all configurations. This consistent performance signifies that FEDEVE is a potent and reliable method for the FEMNIST dataset across the tested configurations.

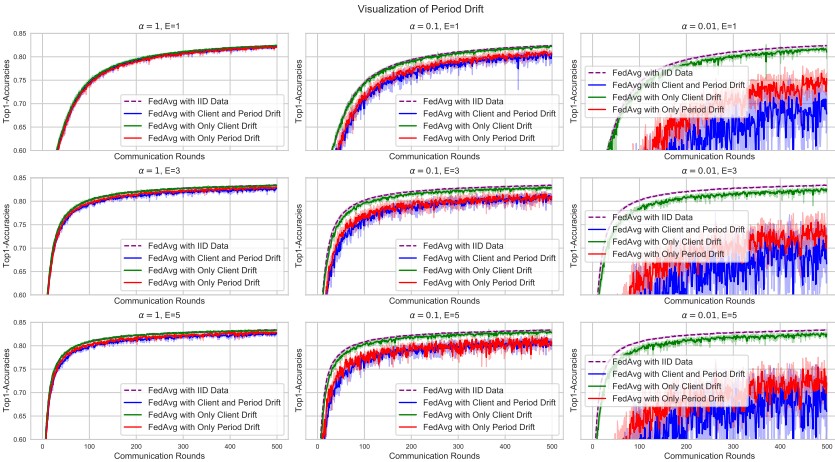

Figure 5: **Visualization of period drift with different $\alpha$ and $E$**

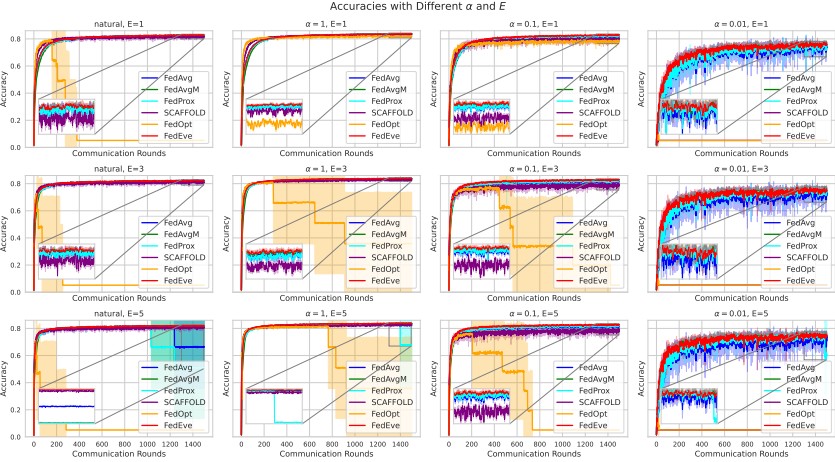

Figure 6: **Accuracies with different $\alpha$ and $E$**

The performance drop of SCAFFOLD in specific experiments may be attributed to two primary reasons: Staleness of control variate: SCAFFOLD mandates that each client maintain a control variant. However, given the large number of clients and the fact that only a limited subset is chosen for training during each communication round, most control variants remain outdated. As a result, they fail to effectively correct the bias in local updates. This point was also reported in FedOpt (Reddi et al., 2020). Excessiveness of correction: Upon detailed inspection of our experiments, we discerned that the training of SCAFFOLD tends to fail when there exists a client with more substantial data than others. This stems from the fact that the fixed batch size and training epoch will result in more local updates in the clients with more data, but it will be corrected by the same control variant in SCAFFOLD. Excessive corrections drive the model further from the optimal point, resulting in the divergence of the model.

We reckon that the poor performances of FedOpt in some settings primarily result from period drift. Period drift impedes FedOpt's adaptivity across rounds. FedOpt tailors the learning rates of individual weights by accumulating past gradients' squares. However, with the ever-shifting optimization objectives in each communication round (period drift), these rate adjustments become misaligned for subsequent updates, thereby skewing model training. It is validated in the experiments that FedOpt fails on FEMNIST with natural and heterogeneity, where period drift is more dominant (more client number, more non-i.i.d. data).

## C GENERALIZATION BOUND

This Generalization bound is inspired by Sun et al. (2023).

### C.1 THEOREMS

**Theorem C.1** (On-average Algorithmic Stability). *Suppose a federated learning algorithm $\mathcal{A}$ is $\epsilon$-on-averagely stable. Then,*

$$\epsilon_{gen} \leq \mathbb{E}_{\mathcal{A}, \mathcal{S}} \left[ \left| R(\mathcal{A}(\mathcal{S})) - \hat{R}_{\mathcal{S}}(\mathcal{A}(\mathcal{S})) \right| \right] \leq \epsilon.$$

**Theorem C.2** (Generalization Bound). *Suppose Assumptions C.3-C.5 hold and $\eta_l \leq \frac{1}{L_m K(t+1)}$. Then,*

$$\epsilon_{gen} \leq \mathcal{O}\left(\frac{L_p}{nL_m}\right) \left[ T\Sigma + \left(\frac{L_m \Delta_0 \Sigma^2}{nK}\right)^{\frac{1}{4}} T^{\frac{3}{4}} + (L_m \Delta_0)^{\frac{1}{2}} T^{\frac{1}{2}} \right].$$

### C.2 ASSUMPTIONS

**Assumption C.3** (Lipschitz Continuity). *The loss function $l(\cdot, z)$ is $L_p$-Lipschitz continous, that is, $|l(w; z) - l(w'; z)| \leq L_p \|w - w'\|$, and is $L_m$-smooth for any $z$, that is, $\|\nabla l(w; z) - \nabla l(w'; z)\| \leq L_m \|w - w'\|$ for any $z, w, w'$.*

**Assumption C.4** (Bounded Variance). *The function $F_i$ have $\sigma_l$-bounded (local) variance i.e., $\mathbb{E}\|g_i(w) - \nabla R_i(w)\| \leq \sigma_l$ for all $w \in \mathbb{R}^d$, $j \in [d]$ and $i \in [m]$. Furthermore, we assume the (global) variance is bounded, $\mathbb{E}\|R_i(w) - \nabla R(w)\| \leq \sigma_g$ for all $x \in \mathbb{R}^d$ and $j \in [d]$.*

**Assumption C.5** (Bounded Gradients). *The function $f_i(x, z)$ have $G$-bounded gradients i.e., for any $i \in [m]$, $x \in \mathbb{R}^d$ and $z \in \mathcal{Z}$ we have $|[\nabla R_i(x, z)]_j| \leq G$ for all $j \in [d]$.*

### C.3 LEMMAS

**Lemma C.6** (Bounded Local Updates). *Suppose Assumptions C.3-C.5 hold. For any step-size, we can bound the local updates as*

$$\mathbb{E}\|w_{i,k} - w_t\| \leq \frac{(1 + \eta_l L_m)^k - 1}{L_m} \left( \mathbb{E}\|\nabla R(w_t)\| + \sigma_l + \sigma_g \right).$$

*where $w_{i,k}$ is the model parameters of client $i$ at $k$-th local updates.*

**Lemma C.7** (Bounded Local Gradients). *Given Assumptions C.3-C.5. For any step-size, we can bound the local gradients as*

$$\mathbb{E}\|g_i(w_{i,k})\| \leq (1 + \eta_l L_m)^k \left( \mathbb{E}\|\nabla R(w_t)\| + \sigma_l + \sigma_g \right),$$

*where $g_i(\cdot)$ is the sampled gradient of client $i$.*

**Lemma C.8** (Bounded Global Model with Sample Perturbation). *Given Assumptions C.3-C.5. For any step-size, we can bound the local gradients as*

$$\mathbb{E}\|w_T - w_T'\| \leq \sum_{t=0}^{T-1} \frac{2e^{\eta_l K(t+1) L_m}}{nL_m} \left( \mathbb{E}\|\nabla R(w_t)\| + \sigma_l + \sigma_g \right),$$

*where $w_T'$ is the model parameters with sample perturbation at $T$-th communication rounds.*

### C.4 PROOFS

#### C.4.1 PROOF OF LEMMA C.6

*Proof.* Bounding Local Updates:

$$\begin{aligned}
\mathbb{E} \quad & \|w_{i,k+1} - w_t\| \\
= \quad & \mathbb{E}\|w_{i,k} - \eta_l g_i(w_{i,k}) - w_t\| \\
\leq \quad & \mathbb{E}\|w_{i,k} - w_t - \eta_l(g_i(w_{i,k}) - g_i(w_t))\| + \eta_l \mathbb{E}\|g_i(w_t)\| \\
\leq \quad & (1 + \eta_l L_m)\mathbb{E}\|w_{i,k} - w_t\| + \eta_l \mathbb{E}\|g_i(w_t)\| \\
\leq \quad & (1 + \eta_l L_m)\mathbb{E}\|w_{i,k} - w_t\| + \eta_l(\mathbb{E}\|g_i(w_t) - \nabla R_i(w_t)\| + \mathbb{E}\|\nabla R_i(w_t) - \nabla R(w_t)\| + \mathbb{E}\|\nabla R(w_t)\|) \\
\leq \quad & (1 + \eta_l L_m)\mathbb{E}\|w_{i,k} - w_t\| + \eta_l(\sigma_l + \sigma_g + \mathbb{E}\|\nabla R(w_t)\|),
\end{aligned}$$

unrolling the above and noting $w_{i,0} = w_t$ yields

$$\mathbb{E}\|w_{i,k} - w_t\| \quad \leq \quad \frac{(1 + \eta_l L_m)^k - 1}{L_m} \left( \mathbb{E}\|\nabla R(w_t)\| + \sigma_l + \sigma_g \right).$$

$\square$

### C.4.2   PROOF OF LEMMA C.7

*Proof.* Bounding Local Gradients:

$$
\begin{aligned}
\mathbb{E}\|g_i(w_{i,k})\| &= \mathbb{E}\|g_i(w_{i,k}) - \nabla R_i(w_{i,k}) + \nabla R_i(w_{i,k}) - \nabla R(w_t) + \nabla R(w_t)\| \\
&\leq \mathbb{E}\|g_i(w_{i,k}) - \nabla R_i(w_{i,k})\| + \mathbb{E}\|\nabla R_i(w_{i,k}) - \nabla R(w_t)\| + \mathbb{E}\|\nabla R(w_t)\| \\
&\leq \sigma_l + L_m \mathbb{E}\|w_{i,k} - w_t\| + \mathbb{E}\|\nabla R(w_t)\|,
\end{aligned}
$$

based on Lemma C.6, we obtain:

$$
\begin{aligned}
&\leq \sigma_l + \mathbb{E}\|\nabla R(w_t) \\
&+ \left((1 + \eta_l L_m)^k - 1\right)\left(\mathbb{E}\|\nabla R(w_t)\| + \sigma_l + \sigma_g\right) \\
&\leq (1 + \eta_l L_m)^k \left(\mathbb{E}\|\nabla R(w_t)\| + \sigma_l + \sigma_g\right).
\end{aligned}
$$

$\square$

### C.4.3   PROOF OF LEMMA C.8

**Definition C.9** (Sample Perturbation). Given a global dataset $\mathcal{S} = \bigcup_{l=1}^{m} \mathcal{S}_l$, where $\mathcal{S}_l$ is the local dataset of the $l$-th client with $\mathcal{S}_l = \{z_{l,1}, \ldots, z_{l,n_l}\}, \forall l \in [m]$, another global dataset is said to be neighboring to $\mathcal{S}$ for client $i$, denoted by $\mathcal{S}^{(i)}$, if $\mathcal{S}^{(i)} := \bigcup_{l \neq i} \mathcal{S}_l \cup \mathcal{S}'_i$, where $\mathcal{S}'_i = \{z_{i,1}, \ldots, z_{i,j-1}, z'_{i,j}, z_{i,j+1}, \ldots, z_{i,n_i}\}$ with $z'_{i,j} \sim P_i$, for some $j \in [n_i]$. And we call $z'_{i,j}$ the perturbed sample in $\mathcal{S}^{(i)}$.

**Definition C.10** (On-average Stability for Federated Learning). A federated learning algorithm $\mathcal{A}$ is said to have $\epsilon$-on-average stability if given any two neighboring datasets $\mathcal{S}$ and $\mathcal{S}^{(i)}$, then

$$
\max_{j \in [n_i]} \mathbb{E}_{\mathcal{A}, \mathcal{S}, z'_{i,j}} |l(\mathcal{A}(\mathcal{S}); z'_{i,j}) - l(\mathcal{A}(\mathcal{S}^{(i)}); z'_{i,j})| \leq \epsilon, \quad \forall i \in [m],
$$

where $z'_{i,j}$ is the perturbed sample in $\mathcal{S}^{(i)}$.

*Proof.* Given time index $t$ and for client $j$ with $j \neq i$, we have

$$
\begin{aligned}
\mathbb{E}\|w_{j,k+1} - w'_{j,k+1}\| &= \mathbb{E}\|w_{j,k} - w'_{j,k} - \eta_l(g_j(w_{j,k}) - g_j(w'_{j,k}))\| \\
&\leq (1 + \eta_l L_m)\mathbb{E}\|w_{j,k} - w'_{j,k}\|.
\end{aligned}
$$

And unrolling it gives

$$
\mathbb{E}\|w_{j,K} - w'_{j,K}\| \leq e^{\eta_l K L_m} \mathbb{E}\|w_t - w'_t\|, \quad \forall j \neq i,
$$

since $1 + x < e^x$. For client $i$, there are two cases to consider. In the first case, SGD selects non-perturbed samples in $\mathcal{S}$ and $\mathcal{S}^{(i)}$, which happens with probability $1 - 1/n_i$. Then, we have

$$
\|w_{i,k+1} - w'_{i,k+1}\| \leq (1 + \eta_l L_m)\|w_{i,k} - w'_{i,k}\|.
$$

In the second case, SGD encounters the perturbed sample at time step $k$, which happens with probability $1/n_i$. Then, we have

$$
\begin{aligned}
\|w_{i,k+1} - w'_{i,k+1}\| &= \|w_{i,k} - w'_{i,k} - \eta_l(g_i(w_{i,k}) - g'_i(w'_{i,k}))\| \\
&\leq \|w_{i,k} - w'_{i,k} - \eta_l(g_i(w_{i,k}) - g_i(w'_{i,k}))\| + \eta_l\|g_i(w'_{i,k}) - g'_i(w'_{i,k})\| \\
&\leq (1 + \eta_l L_m)\|w_{i,k} - w'_{i,k}\| + \eta_l\|g_i(w'_{i,k}) - g'_i(w'_{i,k})\|.
\end{aligned}
$$

Combining these two cases for client $i$ we have

$$
\begin{aligned}
\mathbb{E}\|w_{i,k+1} - w'_{i,k+1}\| &\leq (1 + \eta_l L_m)\mathbb{E}\|w_{i,k} - w'_{i,k}\| + \frac{\eta_l}{n_i}\mathbb{E}\|g_i(w'_{i,k}) - g'_i(w'_{i,k})\| \\
&\leq (1 + \eta_l L_m)\mathbb{E}\|w_{i,k} - w'_{i,k}\| + \frac{2\eta_l}{n_i}\mathbb{E}\|g_i(w_{i,k})\|,
\end{aligned}
$$

based on Lemma C.7, we obtain:

$$
\begin{aligned}
&\leq (1 + \eta_l L_m)\mathbb{E}\|w_{i,k} - w'_{i,k}\| \\
&+ \frac{2\eta_l}{n_i}e^{\eta_l k L_m}\left(\mathbb{E}\|\nabla R(w_t)\| + \sigma_l + \sigma_g\right),
\end{aligned}
$$

then unrolling it gives

$$
\begin{aligned}
\mathbb{E} \quad & \|w_{i,K} - w'_{i,K}\| \\
\leq \quad & e^{\eta_l K L_m} \mathbb{E}\|w_t - w'_t\| \\
+ \quad & \frac{2e^{\eta_l K L_m}}{n_i L_m} \left( \mathbb{E}\|\nabla R(w_t)\| + \sigma_l + \sigma_g \right) \quad \forall j = i.
\end{aligned} \tag{35}
$$

Combines 35 and 35 we have

$$
\begin{aligned}
\mathbb{E}\|w_{t+1} - w'_{t+1}\| \quad \leq \quad & \sum_{i=1}^{m} p_i \mathbb{E}\|w_{i,K} - w'_{i,K}\| \\
\leq \quad & e^{\eta_l K L_m} \mathbb{E}\|w_t - w'_t\| + \frac{2e^{\eta_l K L_m}}{n L_m} \left( \mathbb{E}\|\nabla R(w_t)\| + \sigma_l + \sigma_g \right)
\end{aligned}
$$

where we also use $p_i = n_i/n$ in the last step. Further, unrolling the above over $t$ and noting $w_0 = w'_0$, we obtain

$$
\mathbb{E}\|w_T - w'_T\| \leq \sum_{t=0}^{T-1} \frac{2e^{\eta_l K(t+1) L_m}}{n L_m} \left( \mathbb{E}\|\nabla R(w_t)\| + \sigma_l + \sigma_g \right).
$$

$\square$

### C.4.4 PROOF OF THEOREM C.2

*Proof.* According to the fact that:

$$
\left( \sum_{t=0}^{T-1} \mathbb{E}\|\nabla R(w_t)\| \right)^2 \leq T \sum_{t=0}^{T-1} \left( \mathbb{E}\|\nabla R(w_t)\| \right)^2 \leq T \sum_{t=0}^{T-1} \mathbb{E}\|\nabla R(w_t)\|^2,
$$

where the second inequality follows Jensen's inequality, and the convergence analysis of FedAvg with momentum:

$$
\frac{1}{T} \sum_{r=0}^{T-1} \mathbb{E}\|\nabla R(w_t)\|^2 \leq \mathcal{O} \left( \sqrt{\frac{L_m \Delta_0 \Sigma^2}{nKT}} + \frac{L_m \Delta_0}{T} \right),
$$

where $\Sigma = \sigma_l + \sigma_g$, $\Delta_0 := \mathbb{E}[R(w_0) - R(w^*)]$. The generalization bound is:

$$
\begin{aligned}
\epsilon_{gen} \quad \leq \quad & L_p \mathbb{E}\|w_T - w'_T\| \\
\leq \quad & L_p \sum_{t=0}^{T-1} \frac{2e^{\eta_l K(t+1) L_m}}{n L_m} \left( \mathbb{E}\|\nabla R(w_t)\| + \sigma_l + \sigma_g \right),
\end{aligned}
$$

when $\eta_l < \frac{1}{K(t+1) L_m}$, we obtain:

$$
\leq \quad \mathcal{O}\left( \frac{L_p}{n L_m} \right) \left[ T\Sigma + \left( \frac{L_m \Delta_0 \Sigma^2}{nK} \right)^{\frac{1}{4}} T^{\frac{3}{4}} + (L_m \Delta_0)^{\frac{1}{2}} T^{\frac{1}{2}} \right],
$$

where $\Sigma = \sigma_l + \sigma_g$, $\Delta_0 := \mathbb{E}[R(w_0) - R(w^*)]$. $\square$

