# OpenReview forum: "FedEve: On Bridging the Client Drift and Period Drift for Cross-device Federated Learning"
_ICLR.cc/2024/Conference — ICLR 2024 Conference Withdrawn Submission_

### Official Review · Reviewer_NDKx · 2023-10-23

**Soundness:** 3 good
**Presentation:** 3 good
**Contribution:** 3 good
**Rating:** 6
**Confidence:** 4

**Summary:**

This work investigates the decoupling of classic client drift which is widely studied in previous works, and proposed "period drift" which is less explored. Authors propose a simple method FedEVE based on their predict-observe framework, and extensive experiments support their proposed method's performance.

**Strengths:**

- This work is well organized and written.
- This work proposes a very simple and effective method based on the Bayesian filter (or Kalman filter). The experimental results support their claim.
- The proposed "period drift" concept is good for federatede learning community to  further study. Authors are encouraged to open-source their source codes for FedEvE and other compared methods which helps to broaden the influence of this work.
- Personally I like the analysis of Kalman Gain a lot : )
- One less studied area discusses how to perform FL under noisy labels [1], future studies can explore this area with the light of  authors' proposed framework.

[1] Jiang X, Sun S, Wang Y, et al. Towards federated learning against noisy labels via local self-regularization[C]//Proceedings of the 31st ACM International Conference on Information & Knowledge Management. 2022: 862-873.

**Weaknesses:**

- Since this work is tightly related to the client selection, so the random seeds to conduct experiments on their proposed method and baseline methods should be given to increase the reproducibility.
- The total client number for other datasets (CIFAR10/100) seems not given.

**Questions:**

See above.

---

### Official Review · Reviewer_GeAV · 2023-10-28

**Soundness:** 2 fair
**Presentation:** 2 fair
**Contribution:** 2 fair
**Rating:** 3
**Confidence:** 4

**Summary:**

This paper proposes the period drift, which means that, participating clients at each communication round may exhibit distinct data distribution. Authors claim that it could be more harmful than client drift since the optimization objective shifts with every round. To this end, this paper investigates the interaction between period drift and client drift, finding that period drift can have a particularly detrimental effect on cross-device FL as the degree of data heterogeneity increases. Then, a predict-observe framework and an instantiated method, FEDEVE is proposed, where these two types of drift can compensate each other to mitigate their overall impact.

**Strengths:**

1. Using Bayesian filter to compensate two sources of drift is novel.
2. The connection between server momentum and Kalman Filter is interesting.
3. The paper is written clearly.

**Weaknesses:**

1. The so called ``period drift'' comes from the stochastic sampling of clients. If we see sampling clients as sampling data in SGD, such a period drift also happens during SGD -- each batch of data has distinct data distribution from other batches. Authors should provide a more rigorous definition of period drift and show that how the period drift harms training.
2. The Figure 3 shows the period drift that the sampled data on one client varies across different rounds. This may still be similar to [1], as indicated in related work. Moreover, to address this varying effect, the clients can traverse the whole local dataset using sampling without replacement.
3. Experiment result show little improvements than baselines.

[1] Diurnal or nocturnal? federated learning of multi-branch networks from periodically shifting distributions.

**Questions:**

1. See weakness 2. When clients traverse the whole local dataset using sampling without replacement, does the period drift still happen?
2. As shown in Figure 3, how fedavg_perod_drift_only is drawn? Specifically, how to guarantee that only period drift happens, but client drift not happens?

---

### Official Review · Reviewer_UbMd · 2023-11-11

**Soundness:** 2 fair
**Presentation:** 2 fair
**Contribution:** 2 fair
**Rating:** 3
**Confidence:** 4

**Summary:**

This paper studies the effect of various "drifts" in Federated learning settings. In particular, the paper focuses on period drift, which arises due to partial participation of clients in FL settings. The paper proposes a predict-observe framework and provides an instantiation of the framework, FedEve, to handle these drifts. Experiments are provided to demonstrate the effectiveness of these approaches. While the paper has interesting elements, I have the following primary concerns:

(1) The so-called "period drift" arises in almost all stochastic optimization methods. Of course, this could be severe in FL settings due to higher data heterogeneity but the presentation of the paper is misleading since it is presented as if it is a new concept.

(2) Missing mathematical rigor: It felt like the paper was missing mathematical rigor. For instance, period drift was not defined in the whole paper. The exact definition of it is missing. Furthermore, at places, terms were introduced without proper mathematical definition (e.g. w_server in Assumption 3.1).

(3) Assumptions in the paper are very strong. While the authors tried to provide some vague justification, this does not represent any realistic scenario.  Assumption 3.2 especially looks very strong and I do not believe it happens in practice. Are there any empirical evidence provided to support these Assumptions (which I may have missed)?

(4) The empirical analysis looks fairly weak. The improvement on most datasets seems somewhat small and experiments do not provide any justification for the assumptions made in the paper.

Overall, while the paper has interesting elements, I believe there are severe shortcomings need to be addressed before publication.

**Strengths:**

Refer to summary

**Weaknesses:**

Refer to summary

**Questions:**

Refer to summary